# The Effect of Social Isolation, Loneliness, and Physical Activity on Depressive Symptoms of Older Adults during COVID-19: A Moderated Mediation Analysis

**DOI:** 10.3390/ijerph21010026

**Published:** 2023-12-23

**Authors:** Min Jin Jin, Sung Man Bae

**Affiliations:** 1Division of Liberal Arts, Kongju National University, Gongju 32588, Republic of Korea; jin@kongju.ac.kr; 2Department of Psychology and Psychotherapy, Dankook University, Cheonan 31116, Republic of Korea

**Keywords:** social isolation, loneliness, physical activity, depressive symptoms, older adults, COVID-19

## Abstract

Depressive symptoms have been commonly reported in older adults during the coronavirus disease (COVID-19) pandemic. Social isolation as a result of social distancing to prevent COVID-19 was reported to increase the level of depressive symptoms both directly and indirectly through the mediating effect of loneliness. Depressive symptoms in older adults can be regulated through health behaviors such as physical activity. Therefore, this study used a moderated mediation model to explain depressive symptoms. The English Longitudinal Study of Ageing COVID-19 wave 2 data were used. A total of 296 individuals were excluded due to missing values, leaving a final sample of 6499. Depressive symptoms, social isolation, loneliness, and physical activity were used in the moderated mediation analysis with various demographic and general health variables as covariates. Loneliness was found to significantly mediate the relationship between social isolation and depression. While moderate physical activity significantly moderated the effects of social isolation and loneliness on depressive symptoms, mild physical activity alone significantly moderated the effects of loneliness on depressive symptoms. This study revealed the impact of social isolation on depressive symptoms directly and indirectly mediated by loneliness, with a moderating effect of moderate and mild physical activity in the elderly during COVID-19 in a moderated mediation model.

## 1. Introduction

Since the coronavirus disease (COVID-19) was first reported in December 2019, the pandemic has posed novel risks to individuals globally in all age groups. In particular, the risk has been reported to be much higher in older adults, both physically and psychologically [1]. One of the most common psychological problems faced by older adults during the COVID-19 pandemic was depression, with 12.8% to 40.1% of older adults reporting experiencing depressive symptoms [2,3,4,5]. Depression among older adults should draw careful attention because it is significantly related to many other psychological problems and suicide risk [3,6,7,8].

Depressive symptoms experienced during the COVID-19 pandemic have been reported to be caused by social isolation in older adults. Studies found that social isolation as a result of social distancing, which was a major strategy to prevent COVID-19, increased the level of depressive symptoms in older adults [4,9]. Given that the elderly usually rely on face-to-face communication and lack the skills to use electronic devices [10], the impact of social isolation caused by the lockdown could block more chances of receiving social support from others.

Depressive symptoms can be indirectly affected by social isolation, as well as directly through the mediating effect of loneliness in older adults [4]. Social isolation and loneliness are considered distinct from each other: while social isolation quantitatively represents the objective lack of social contact, loneliness is considered a subjective feeling of a perceived deficiency in social relationships [11,12]. As they are different constructs, they represent closely related but different risk pathways for depression [13]. A recent study found that COVID-19 social isolation that did not pre-exist increased loneliness, which later demonstrated a significant relationship with depression in older adults [4]. Social isolation due to social distancing seems to cause loneliness, which could be a risk factor for depression [14]. This loneliness–depression association is especially strong in those with an older age identity [15].

Meanwhile, depressive symptoms associated with social isolation and loneliness in the elderly can be regulated through health-related behaviors such as physical activity [16]. Previous studies have reported that physical activity predicted depression in older adults during the COVID-19 pandemic [17,18]. Daily moderate and moderate-to-vigorous physical activity were negatively associated with depression among older adults during the COVID-19 pandemic [19]. In addition, physical activity was a significantly relevant variable regarding the relationship between social activity and depression [17,20] and between loneliness and depression [21]. Thus, the effect of physical activity can buffer the effects of social isolation and loneliness on depression. However, studies examining the direct and indirect effects of social isolation through the mediation of loneliness and the moderating effect of physical activity on depressive symptoms in older adults using a comprehensive model have not yet been conducted. 

Therefore, this study aimed to examine a moderated mediation model to explain depressive symptoms in older adults during the COVID-19 pandemic. In particular, both the mediating effect of loneliness on the relationship between social isolation and depressive symptoms and the moderating effect of physical activity on depressive symptoms were investigated using a moderated mediation model. To our knowledge, this is the first study examining a comprehensive moderated mediation model to provide an understanding of the development and intervention for depressive symptoms during the COVID-19 pandemic in older adults. The hypotheses of this study were the following: (1) the higher social isolation score would be associated with a high level of depressive symptoms (direct path); (2) the social isolation score would be associated with a higher loneliness score, and the higher loneliness score would be associated with a higher level of depressive symptoms (indirect path of mediation); (3) physical activity would moderate effects of social isolation and loneliness on depressive symptoms, which could mean that individuals with a higher level of physical activity would experience lower levels of depressive symptoms (moderated mediation); and (4) the moderating effect of physical activity would be different based on the severity of activity.

## 2. Materials and Methods

The data used in this study were collected during the COVID-19 pandemic as part of the English Longitudinal Study of Ageing (ELSA), which is an ongoing national cohort study of older adults 50 years and older living in England [22]. The ELSA COVID-19 wave 2 data, which were collected in November and December 2020, were the latest among all the ELSA data. A mix of Internet- and phone-based questionnaires with cognitive and physical examinations and self-reported questionnaires were used. The ELSA COVID-19 Wave 2 data consist of 6794 individuals in total. 

Psychological variables used in this study included depressive symptoms, social isolation, loneliness, and physical activity. The following variables were used from the datasets to control for possible contaminating effects according to previous studies [23,24]: demographic variables included gender, age, ethnicity, relationship status, and tenure as an index of socioeconomic status, and general health variables included smoking; weight; height; body mass index (BMI); perceived health status of the last month; perceived sleep status of the last month; and having long-standing illness, disability, or infirmity. A total of 296 individuals were excluded due to missing values for key variables, leaving a final sample of 6499. The final sample consisted of 2895 (44.5%) men and 3604 (55.5%) women, with a mean age of 68.66 years (standard deviation, SD = 9.52). 

Depressive symptoms were assessed using the Center for Epidemiological Studies Depression (CES-D-8) scale [25]. The CES-D-8 consists of 8 items with a dichotomous response format, including questions such as “Much of the time during the past week: You felt depressed” and “Much of the time during the past week: felt sad”, for example. We coded depressive answers as 1 and non-depressive answers as 0, which made a score ranging from 0 to 8, with a high score representing more depressive symptoms. In this study, Cronbach’s α for internal consistency reliability was 0.835.

Social isolation during the COVID-19 pandemic was ascertained using two items: self-isolation in the previous week and trying to stay at home during the last week. These items were assessed using yes or no binary responses. We coded “yes” as 1 and “no” as 0, which made a score range between 0 and 2, with a high value representing the more socially isolated experience.

Loneliness was assessed using the University of California, Los Angeles Loneliness Scale (Short Form) (ULS-4) [26]. The ULS-4 consists of 4 items, including questions such as “How often do you feel you lack companionship?” and “How often do you feel lonely?”, for example. It is assessed on a 3-point Likert scale ranging from 1 (“hardly ever or never”) to 3 (“often”). The total score ranges from 1 to 12, with a high score indicating greater loneliness. Cronbach’s α for internal consistency reliability in this study was 0.875.

Physical activity was ascertained using 3 items on the frequency of participation in vigorous, moderate, and mild activities. These items were assessed on a 4-point Likert scale ranging from 1 (“hardly ever or never”) to 4 (“more than once a week”).

Demographic variables and general health variables from the existing datasets were set as covariates including gender (1 = male; 2 = female), age (years), ethnicity (1 = Any White; 2 = Black, Asian, and minority ethnic), relationship status (1 = Married; 2 = In a registered civil partnership; 3 = Living with a partner; 4 = With a partner you do not live with; 5 = Separated (after being married or in a civil partnership); 6 = Divorced/dissolved civil partnership; 7 = Widowed/surviving partner from a civil partnership; 8 = Single), tenure (1 = Own it outright; 2 = Buying it with the help of a mortgage or loan; 3 = Pay part rent and part mortgage (shared ownership); 4 = Rent it; 5 = Live here rent free (including rent free in relative’s/friend’s property; excluding squatting); 6 = Squatting), smoking (1 = Yes; 2 = No), weight (in kilograms), height (in centimeters), perceived health status of the last month (1 = Excellent; 2 = Very good; 3 = Good; 4 = Fair; 5 = Poor), perceived sleep status of the last month (1 = Excellent; 2 = Very good; 3 = Good; 4 = Fair; 5 = Poor), and having long-standing illness, disability, or infirmity (1 = Yes; 2 = No).

The normality of the key variables was tested using skewness and kurtosis. A correlation analysis was performed to investigate the possible contaminating effects of demographic and general health variables on depressive symptoms. After setting the covariates, partial correlation and regression analyses were performed to discover relationships among the variables. The significance level was set at *p* < 0.05 (two-tailed). Statistical analyses were performed using SPSS 26 (SPSS, Inc., Chicago, IL, USA), and regression analyses were performed using SPSS Macro PROCESS software SPSS 4.0 [27] to examine the moderated mediating effect. 

All analyses were performed in accordance with the guidelines and regulations of the Institutional Review Board of Dankook University (IRB no. 2023-11-002).

## 3. Results

### 3.1. Descriptive Data and Correlations

All variables in our results were within the normal distribution range, with skewness less than 2.0 and kurtosis less than 7.0 [28]. Pearson’s correlation analysis was performed to investigate the possible contaminating effects of demographic and general health variables on depressive symptoms. While gender (*r* = 0.158, *p* < 0.001; significant correlation between female and more depressive symptom), age (*r* = −0.047, *p* < 0.001; significant correlation between lesser age and more depressive symptom), relationship status (*r* = 0.160, *p* < 0.001), tenure (*r* = 0.147, *p* < 0.001), smoking (*r* = −0.067, *p* < 0.001), weight (*r* = 0.052, *p* < 0.001), height (*r* = −0.122, *p* < 0.001), BMI (*r* = 0.131, *p* < 0.001), the perceived health status of the last month (*r* = 0.459, *p* < 0.001), perceived sleep status of the last month (*r* = 0.523, *p* < 0.001), and having long-standing illness, disability, or infirmity (*r* = −0.229, *p* < 0.001) were significantly correlated with depressive symptoms, ethnicity was not (*r* = 0.007, *p* = 0.594). Therefore, the following were controlled for as covariates in further analyses: sex; age; relationship status; tenure; smoking; weight; height; BMI; perceived health status in the previous month; perceived sleep status in the last month; and long-standing illness, disability, or infirmity.

A partial correlation analysis was performed to determine the relationships among depressive symptoms, social isolation, loneliness, and physical activity. Table 1 presents the means, standard deviations, and correlation coefficients of the variables. Depressive symptoms were significantly correlated with social isolation (*r* = 0.051, *p* < 0.001), loneliness (*r* = 0.526, *p* < 0.001), moderately energetic physical activity (*r* = −0.042, *p* = 0.001), and mildly energetic physical activity (*r* = −0.044, *p* < 0.001) but not with vigorous physical activity (*r* = −0.007, *p* = 0.597). Social isolation was significantly correlated with loneliness (*r* = 0.051, *p* < 0.001) and vigorous physical activity (*r* = −0.037, *p* = 0.003) but not with moderate physical activity (*r* = −0.016, *p* = 0.210) or mildly energetic physical activity (*r* = 0.014, *p* = 0.261). Loneliness was not significantly correlated with vigorous (*r* = 0.015, *p* = 0.216), moderate (*r* = −0.004, *p* = 0.746), or mild physical activity (*r* = −0.003, *p* = 0.831). Therefore, moderate and mild physical activities could be set as possible moderating variables because they showed a significant correlation with the dependent variable, whereas they did not show a significant correlation with the independent or mediating variables. However, vigorous physical activity was not set as a possible moderating variable because it did not show a significant correlation with the dependent variable. 

### 3.2. Mediation Analysis

A mediation model from PROCESS macro number 4 was used to determine the mediating effect of loneliness on the relationship between social isolation and depressive symptoms.

The direct path from social isolation to depressive symptoms was also significant (*B* = 0.106, *p* = 0.015). Paths from social isolation to loneliness (*B* = 0.184, *p* < 0.001) and from loneliness to depressive symptoms (*B* = 0.552, *p* < 0.001) were also significant. The direct effect of social isolation on depressive symptoms was significant (*β* = 0.156, CI [0.021, 0.191]), and the indirect (mediated) effect of social isolation on depressive symptoms was also significant (*β* = 0.102, CI [0.053, 0.155]). Therefore, loneliness significantly mediates the relationship between social isolation and depressive symptoms.

### 3.3. Moderated Mediation Analysis

A moderated mediation model from PROCESS macro number 15 was used to discover the mediating effect of loneliness on the relationship between social isolation and depressive symptoms and the moderating effect of moderate and mild physical activities on both social isolation and loneliness on depressive symptoms.

Table 2 shows the results of the moderated mediation analysis with the moderating effect of moderate physical activity. The direct path from social isolation to depressive symptoms was significant (*B* = 0.428, *p* = 0.001). Paths from social isolation to loneliness (*B* = 0.184, *p* < 0.001) and from loneliness to depressive symptoms (*B* = 0.614, *p* < 0.001) were significant as well. Meanwhile, moderate physical activity significantly moderated the relationship between social isolation and depressive symptoms (*B* = −0.099, *p* = 0.008) and the relationship between loneliness and depressive symptoms (*B* = −0.021, *p* = 0.009). The mediated moderation index (*β* = −0.004, CI [−0.008, −0.001]) and moderation effects were significant, given that the R^2^ was increased due to both interactions (Δ*R*^2^ = 0.001, Δ*F* = 7.085, *p* = 0.008; Δ*R*^2^ = 0.001, ΔF = 6.846, *p* = 0.009, respectively). As revealed by the Johnson-Neyman technique [29], moderate physical activity would moderate the direct effect of social isolation on depressive symptoms when a moderate physical activity score was lower than 3.445, in which the 95% CI did not contain zero. At the same time, moderate physical activity moderated the effect of loneliness on depressive symptoms throughout all scores. Figure 1 shows the moderating effect of moderate physical activity. 

Table 3 shows the results of the moderated mediation analysis with the moderating effect of mild physical activity. The direct path from social isolation to depressive symptoms was significant (*B* = 0.054, *p* = 0.725). However, paths from social isolation to loneliness (*B* = 0.184, *p* < 0.001) and from loneliness to depressive symptoms (*B* = 0.624, *p* < 0.001) were significant. Mild physical activity significantly moderated the relationship between loneliness and depressive symptoms (*B* = −0.022, *p* = 0.018) but not the relationship between social isolation and depressive symptoms (*B* = −0.016, *p* = 0.701). This mediated moderation index was significant (*β* = −0.004, CI [−0.009, −0.001]), and the moderation effect was significant, given that the R^2^ was increased due to the interaction (Δ*R*^2^ = 0.001, Δ*F* = 5.556, *p* = 0.018). The result of the Johnson–Neyman technique showed that mild physical activity moderated the effect of loneliness on depressive symptoms throughout all scores. Figure 2 shows the moderating effect of mild physical activity.

## 4. Discussion

This study investigated a moderated mediation model to explain depressive symptoms among older adults during the COVID-19 pandemic. This study examined the effects of social isolation due to social distancing on depressive symptoms and the mediating effect of loneliness in older adults. Additionally, the moderating effects of moderate and mild physical activity on depressive symptoms were influenced by social isolation and loneliness in older adults. 

The results revealed that the impact of social isolation on depressive symptoms is directly and indirectly mediated by loneliness in older adults. This is in line with previous studies that have explored the effects of social isolation as well as loneliness on depression in older adults [30,31,32,33,34,35]. While social isolation and loneliness are considered different from each other [11,36], they show significant relationships with each other and depression in older adults [37,38]. Studies have revealed that social isolation is significantly associated with loneliness later in time [39] and that loneliness resulting from social isolation has a significant relationship with depression in older adults [4,14]. This study also adds further evidence that social isolation, which was assessed as the objective social distancing due to the lockdown, would not only impact feelings of loneliness but also depressive symptoms in older adults during COVID-19. 

This study also identified the effects of physical activity on the mediation model in older adults. Moderate physical activity significantly moderated the effects of social isolation and loneliness on depressive symptoms. Social isolation was associated with an increase in depressive symptoms only in people with less moderate physical activity; however, social isolation did not show a significant association with depressive symptoms in people with moderate physical activity daily. Similarly, loneliness showed a steeper relationship with depressive symptoms in those who engaged in less than moderate physical activity. Mild physical activity, on the other hand, significantly moderated the effect of loneliness on depressive symptoms only. Loneliness showed a steeper relationship with depressive symptoms in those who engaged in less than moderate physical activity. These moderated mediation results suggest a moderating role for moderate or mild physical activity in decreasing depressive symptoms in older adults. These results revealed the moderating effect of physical activity on depression, which is in line with previous studies showing a significant relationship between physical activity and depression and the moderating effect of physical activity on the relationship between social isolation and depressive symptoms [16,40,41].

Physical activity is not only known as an evidence-based treatment for depression [42] but is also known to be related to behavioral activation therapy for depression, which induces behavioral attitudes and healthy behaviors [43]. Behavioral activation is targeted at reducing behavioral inertia and avoidance, which often accompanies depression, by increasing daily activities [44]. A recent study also suggested that physical activity is an important component and logical basis of behavioral activation interventions for depression [45]. Opportunities to engage in physical activity during the COVID-19 pandemic were impaired due to the lockdown and financial insecurities [18]. That is, engaging in physical activity regularly during the pandemic might establish individuals’ health competence and utilize physical activity as a coping strategy [46,47]. Therefore, improving physical activity could act as a potential therapeutic component of behavioral activation for interventions in depressive symptoms.

On the other hand, results showed an insignificant relationship between physical activity and social isolation and loneliness, which supports previous studies that physical activity did not show a significant relationship with social isolation or loneliness [48,49,50]. It was possible to set physical activity as the moderator in this study since moderating variables should not correlated with independent variables [51]. Although some studies have reported contradictory outcomes suggesting a significant relationship between physical activity, social isolation, and loneliness [17,52,53], not all studies have found a direct negative relationship between physical activity and social isolation or loneliness, which raises the possibility of other moderators and mediators within this relationship [52]. Thus, future studies should include other underlying variables to identify the relationship between physical activity and social isolation or loneliness with careful consideration of possible mediators or moderators.

This study is significant because it explored depressive symptoms using a moderated mediation model of social isolation, loneliness, and physical activity in older adults during the COVID-19 period. Previous studies focused on discovering relationships among social isolation, loneliness, and depressive symptoms [23,54,55], particularly the mediating effect of loneliness between social isolation and depressive symptoms [4,14,15]. However, there is no study yet to discover the possible moderation effect of physical activity with the mediating effect. To our knowledge, this study is the first study to examine the comprehensive moderated mediation model of social isolation, loneliness, depressive symptoms, and physical activity altogether in older adults. The results of this study provide evidence for the knowledge of depressive symptoms, with the activating effect of social isolation directly and indirectly mediated by loneliness and with the buffering effect of moderate and mild physical activities of older adults during a pandemic. While COVID-19 has become endemic, other infectious diseases may emerge and cause significant adverse effects on mental health, such as Middle East respiratory syndrome (MERS) and severe acute respiratory syndrome (SARS), which came before COVID-19. MERS, SARS, and COVID-19 share similar features, including highly contagious zoonotic viruses [56], and are all related to mental health problems, including depression [57,58]. The results of this study could serve as a stepping stone to understanding the activating and buffering effects of depression in the elderly during the next possible pandemic. In particular, the results of this study would provide support for decreasing social isolation and loneliness by increasing physical activity as an intervention for depression.

These results were found while controlling for the contaminating effect of demographic variables, including sex, age, relationship status, and tenure, and general health variables, including smoking, weight, height, BMI, perceived health status in the last month, perceived sleep status in the last month, and long-standing illness, disability, or infirmity. Although this study found significant relationships between depressive symptoms and each covariate, it examined the independent effects of social isolation, loneliness, and physical activity on depression for internal validity. As a result, this study provides a general understanding of depression among older adults, regardless of demographic or health differences. Further studies with more targeted samples are needed to broaden our knowledge.

Despite its significance, this study has some limitations. First, it operationally defines social isolation as an objective state of social distancing, such as self-isolation and trying to stay at home during the COVID-19 pandemic. However, social isolation could include other perspectives, such as the absence of contact with other people or even a subjective feeling of lack of belongingness [59,60]. Although this study focused on social isolation caused by social distancing due to the pandemic, further studies including various components of social isolation and comparing them before and after the COVID-19 pandemic should be conducted to improve our understanding. Moreover, this study was performed with a cross-sectional design using wave 2 data of ELSA COVID-19 alone. This design was internally made since wave 2 data were collected in November and December 2020, while wave 1 data were collected in June and July 2020, the earlier stage of the pandemic, and might not reflect the problems caused by the COVID-19 lockdown. Further longitudinal studies should be conducted after releasing new data. In addition, some correlation coefficients and mediated moderation indices were statistically significant but weak due to the large sample size. However, the large sample size resembles the true population [61], so coefficients fluctuate less, which makes results meaningful enough. In fact, the weak but significant results of this study were in line with previous studies. To discover the exact and reliable results, further studies with smaller samples could reduce the possible error regarding the sample size. Finally, this study was conducted using a general sample from the elderly population in England. Additional studies with other populations from various cultures would be necessary to generalize our findings. In addition, separate studies of patients with depression and healthy populations should be conducted to establish a deeper understanding of depressive symptoms in older adults.

## 5. Conclusions

In conclusion, the results suggest a moderated mediation model that examines the direct and indirect effects of social isolation through the mediating effect of loneliness and the moderating effect of physical activity on depressive symptoms in older adults during the COVID-19 pandemic. Moderate physical activity significantly moderated the effects of social isolation and loneliness on depressive symptoms, whereas mild physical activity only significantly moderated the effects of loneliness on depressive symptoms. The results of this study are in line with those of previous studies and provide evidence for interventions for depression, which could lead to decreasing social isolation or loneliness and increasing physical activity.

## Figures and Tables

**Figure 1 ijerph-21-00026-f001:**
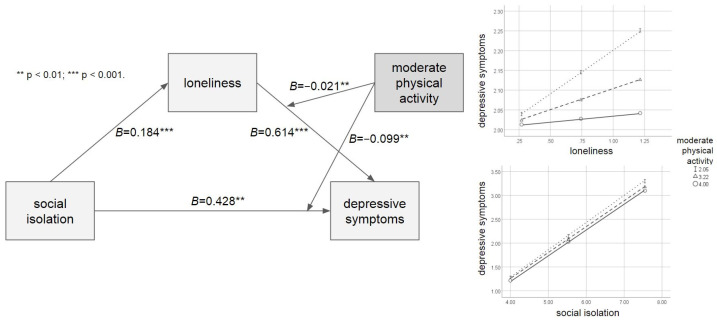
The moderated mediation model with the moderating effect of moderate physical activity.

**Figure 2 ijerph-21-00026-f002:**
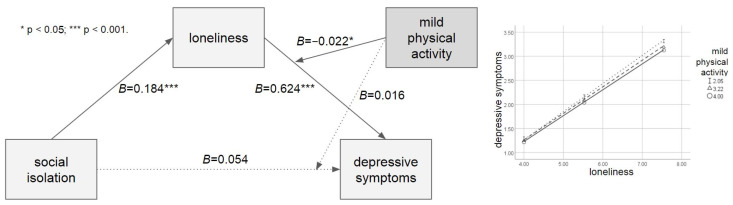
The moderated mediation model with the moderating effect of mild physical activity.

**Table 1 ijerph-21-00026-t001:** Mean, standard deviation, and correlation coefficients of variables.

	1	2	3	4	5	6
1. depressive symptoms	-					
2. social isolation	0.051 ***	-				
3. loneliness	0.526 ***	0.047 ***	-			
4. vigorous physical activity	−0.007	−0.037 **	0.015	-		
5. moderate physical activity	−0.042 **	−0.016	−0.004	0.247 ***	-	
6. mild physical activity	−0.044 ***	0.014	0.003	0.146 ***	0.434 ***	-
**M**	2.086	0.743	5.530	2.010	3.216	3.458
**SD**	2.339	0.472	2.018	1.249	1.163	0.986

Note: N = 6499; ** *p* < 0.01; *** *p* < 0.001.

**Table 2 ijerph-21-00026-t002:** The result of moderated mediation analysis with the moderating effect of moderate physical activity.

	*B*	*SE*	*t*		95% CI	
					LLCI	ULCI
DV: loneliness (*R* = 0.475, *R*^2^ = 0.225, MSE = 3.161, *F* = 171.560 ***)						
social isolation	0.184	0.048		3.808 ***	0.089	0.279
DV depressive symptoms (*R* = 0.735, *R*^2^ = 0.540, MSE = 2.535, *F* = 506.441 ***)						
social isolation	0.428	0.130		3.306 **	0.174	0.682
loneliness	0.614	0.026		23.551 ***	0.563	0.665
moderate physical activity	0.128	0.056		2.289 *	0.018	0.237
social isolation × moderate physical activity	−0.099	0.037		−2.662 **	−0.173	−0.026
loneliness × moderate physical activity	−0.021	0.008		−2.615 **	−0.036	−0.005
Index of moderated mediation	*SE*				95% CI	
					LLCI	ULCI
−0.004	0.002				−0.008	−0.001

Note: DV: dependent variable; SE: standard error; CI: confidence interval; LLCI: lower limit confidence interval; ULCI: upper limit confidence interval; MSE: mean squared error. N = 6499; * *p* < 0.05; ** *p* < 0.01; *** *p* < 0.001.

**Table 3 ijerph-21-00026-t003:** The result of moderated mediation analysis with the moderating effect of mild physical activity.

	*B*	*SE*	*t*		95% CI	
					LLCI	ULCI
DV: loneliness (*R* = 0.475, *R*^2^ = 0.225, MSE = 3.161, *F* = 171.560 ***)						
social isolation	0.184	0.048		3.808 ***	0.089	0.279
DV depressive symptoms (*R* = 0.734, *R*^2^ = 0.539, MSE = 2.527, *F* = 505.533 ***)						
social isolation	0.054	0.152		0.353	0.174	0. 682
loneliness	0.624	0.032		19.244 ***	0.563	0.665
mild physical activity	0.025	0.064		0.385	0.018	0.237
social isolation × mild physical activity	0.016	0.042		0.384	−0.173	−0.026
loneliness × mild physical activity	−0.022	0.009		−2.357 *	−0.036	−0.005
Index of moderated mediation	*SE*				95% CI	
					LLCI	ULCI
−0.004	0.002				−0.009	−0.001

Note: DV: dependent variable; SE: standard error; CI: confidence interval; LLCI: lower limit confidence interval; ULCI: upper limit confidence interval; MSE: mean squared error. N = 6499; * *p* < 0.05; *** *p* < 0.001.

## Data Availability

https://www.elsa-project.ac.uk/covid-19-data, accessed on 18 November 2023.

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
