# Peer review of "The Effect of Social Isolation, Loneliness, and Physical Activity on Depressive Symptoms of Older Adults during COVID-19: A Moderated Mediation Analysis"

_ijerph, 2023, doi:10.3390/ijerph21010026_

Round 1

Reviewer 1 Report

Comments and Suggestions for Authors

The paper titled "Effect of Social Isolation Loneliness and Physical Activity on Depressive Symptoms of Older Adults During COVID-19: A Moderated-Mediation Analysis" provides a thorough investigation into the psychological impacts of the COVID-19 pandemic on older adults. The authors use a moderated-mediation model to assess how social isolation and loneliness contribute to depressive symptoms, and how physical activity can modulate these effects.

Abstract

·       The following sentence is unclear: “While social isolation as a result of social distancing to 11 prevent COVID‐19 was reported to increase depressive symptoms, social isolation seemed to cause 12 loneliness that later affect depression”.

Introduction

·       The following sentence needs a citation: “Depressive symptoms can be indirectly affected by social isolation, as well as directly through the mediating effect of loneliness.”

·       The investigation into the modality of physical activity primarily focuses on the intensity level, while the individual versus group dynamic remains unexamined. This omission may lead to a confounding association, given that group/individual-based physical activities could concurrently influence social isolation and feelings of loneliness.

·       The theoretical underpinnings of the study warrant a more comprehensive elaboration, necessitating the incorporation of established reference theoretical models to bolster the foundational premise and to strengthen the rationale for the current study.

·       To ensure the robustness and applicability of the study's findings, it is imperative to address the concern that the research investigates a highly specialized demographic, which could potentially possess distinctive characteristics. This necessitates a careful consideration of how these unique attributes may influence the study's generalizability and the interpretation of its outcomes. The literature review should also be focused on such aspects. The paper needs to demonstrate a comprehensive understanding of all relevant research to justify its study.

·       The paper indicates that depressive symptoms are a significant issue for older adults during the COVID-19 pandemic due to increased social isolation. A potential issue might be whether the paper sufficiently establishes the novelty of its contribution to the existing literature.

·       The paper outlines its main objectives, but it might be critiqued for not stating specific hypotheses or research questions, which could provide clarity and focus for the reader.

Materials and methods

·       Please provide an example of item for the scales adopted in the study.

·       Please specify direction of the correlation between gender and depression.

·       Please describe the nature of the relationship between age and depression.

·       Could you elucidate on the coding schema employed for the variable of ethnicity? Is it not classified as a categorical variable?

·       Many of the correlations denoted as significant may be exaggerated due to the sample size. The magnitude of the effect size for the associations between depressive symptoms and factors such as social isolation, loneliness, moderately energetic physical activity, and mildly energetic physical activity appears to be minimal.

·       Social isolation was ascertained using two binary response items. This operational definition may not capture the full spectrum of social isolation experiences

·       The sentence implies that the analyses are driven by data rather than by theory, as would be appropriate. “Therefore, moderate and mild physical activities could be set as possible moderating variables because they showed a significant correlation with the dependent variable, whereas they did not show a significant correlation with the independent or mediating variables. However, vigorous physical activity was not set as a possible moderating variable because it did not show a significant correlation with the dependent variable.”

Results

·       Upon examination of the correlation matrix, the sole discernible significant effect appears to be the positive association between loneliness and depression, alongside those among the different modalities of physical activity.

·       Please provide the beta coefficients for the regression models to ascertain the magnitude of the effect.

·       Why was Model 15 employed instead of other models, such as Model 7 or 8? Please explicate the theoretical rationale underpinning this decision.

·       The mediated moderation index (β = -.004, CI [-.008, -.001]) and moderation effects were reported as significant. However, these effects appear to be extremely small.

·       Please, verify the accuracy of the reported R².

·       The study controls for several covariates, but it is not clear how the selection of these covariates was determined, or whether additional variables could have influenced the results.

·       The paper excluded individuals with missing values, which could introduce bias if the missing data are not random.

Discussion

·       The study defines social isolation in terms of social distancing behaviors like self-isolation and staying at home. This definition may not capture the full extent or impact of social isolation on mental health, which can include aspects such as the absence of meaningful social relationships or a subjective feeling of loneliness.

·       The study’s cross-sectional design limits its ability to establish causality between the variables of interest. Longitudinal studies would be more informative in terms of how changes over time in social isolation, loneliness, and physical activity relate to changes in depressive symptoms.

·       The study's findings are based on data from older adults in England during a specific period of the COVID-19 pandemic. These findings may not be generalizable to other populations, times, or contexts.

·       The study concludes that moderate to mild physical activity can reduce depressive symptoms influenced by social isolation and loneliness. However, the discussion acknowledges that physical activity does not have a significant relationship with social isolation or loneliness. This might suggest the need for exploring other factors that could mediate or moderate the relationship between these variables.

·       The study discusses the potential applicability of its findings to future pandemics. While this is an important consideration, the unique characteristics of each pandemic situation need to be taken into account when applying these findings.

·       Please, may the authors elucidate how these findings correspond with those derived from the subsequent paper, and identify the elements that theoretically and empirically distinguish the two studies. https://www.mdpi.com/1660-4601/20/19/6900

Comments on the Quality of English Language

The overall quality of the English is sufficient, with only a few minor issues that need checking.

Author Response

Reviewer #1

The paper titled "Effect of Social Isolation Loneliness and Physical Activity on Depressive Symptoms of Older Adults During COVID-19: A Moderated-Mediation Analysis" provides a thorough investigation into the psychological impacts of the COVID-19 pandemic on older adults. The authors use a moderated-mediation model to assess how social isolation and loneliness contribute to depressive symptoms, and how physical activity can modulate these effects.

Abstract

Question 1: The following sentence is unclear: “While social isolation as a result of social distancing to 11 prevent COVID‐19 was reported to increase depressive symptoms, social isolation seemed to cause 12 loneliness that later affect depression”.

Answer 1: Thank you for your comments. I revised the sentence more clearly: “Social isolation as a result of social distancing to prevent COVID‐19 was reported to increase the level of depressive symptoms both directly and indirectly through the mediating effect of loneliness”.

before

after

While social isolation as a result of social distancing to prevent COVID‐19 was reported to increase depressive symptoms, social isolation seemed to cause loneliness that later affect depression.

Social isolation as a result of social distancing to prevent COVID‐19 was reported to increase the level of depressive symptoms both directly and indirectly through the mediating effect of loneliness.

Introduction

      Question 2: The following sentence needs a citation: “Depressive symptoms can be indirectly affected by social isolation, as well as directly through the mediating effect of loneliness.”

   Answer 2: Thank you for your comments. I added a citation following to the sentence.

before

after

Depressive symptoms can be indirectly affected by social isolation, as well as directly through the mediating effect of loneliness.

Depressive symptoms can be indirectly affected by social isolation, as well as directly through the mediating effect of loneliness in older adults [4].

      Question 3: The investigation into the modality of physical activity primarily focuses on the intensity level, while the individual versus group dynamic remains unexamined. This omission may lead to a confounding association, given that group/individual-based physical activities could concurrently influence social isolation and feelings of loneliness.

   Answer 3: I appreciate your keen advice. Most of the reference cited in the paper is focused on individuals’ personal physical activity that was self-reported (e.g., Callow et al., 2020; Siegmund, et al., 2021; Lage et al., 2021). This study also focused on individuals’ personal physical activity that was self-reported in the ELSA COVID-19 dataset. This individuals’ personal physical activity would be considered as healthy act or habit that could establish individuals' health competence and utilized as a coping strategy. Therefore, this personal physical activity would not interfere with influence social isolation and feelings of loneliness. 

     Question 4: The theoretical underpinnings of the study warrant a more comprehensive elaboration, necessitating the incorporation of established reference theoretical models to bolster the foundational premise and to strengthen the rationale for the current study.

   Answer 4: Thank you for your advice. Previous studies reported the mediation model of loneliness between social isolation and depressive symptoms (Banerjee, 2020; Robb et. al., 2020; Shrira et al., 2020). However, there is no study yet to discover this mediation effect with possible moderation effect. Therefore, this study suggested the moderating effect of physical activity on depressive symptoms. This moderated mediation model is comprehensive yet simple enough that it was established from previous studies which I have already cited in the paper.

Banerjee, D. (2020). The impact of Covid‐19 pandemic on elderly mental health. International journal of geriatric psychiatry, 35(12), 1466

Robb, C. E., De Jager, C. A., Ahmadi-Abhari, S., Giannakopoulou, P., Udeh-Momoh, C., McKeand, J., ... & Middleton, L. (2020). Associations of social isolation with anxiety and depression during the early COVID-19 pandemic: a survey of older adults in London, UK. Frontiers in psychiatry, 11, 591120.

Shrira, A., Hoffman, Y., Bodner, E., & Palgi, Y. (2020). COVID-19-related loneliness and psychiatric symptoms among older adults: the buffering role of subjective age. The American Journal of Geriatric Psychiatry, 28(11), 1200-1204.

before

after

This comprehensive model could provide an understanding of the development and intervention for depressive symptoms during the COVID-19 pandemic in older adults.

To our knowledge, this is the first study examining a comprehensive moderated mediation model to provide an understanding of the development and intervention for depressive symptoms during the COVID-19 pandemic in older adults.

      Question 5: To ensure the robustness and applicability of the study's findings, it is imperative to address the concern that the research investigates a highly specialized demographic, which could potentially possess distinctive characteristics. This necessitates a careful consideration of how these unique attributes may influence the study's generalizability and the interpretation of its outcomes. The literature review should also be focused on such aspects. The paper needs to demonstrate a comprehensive understanding of all relevant research to justify its study.

   Answer 5: I understand your concern. I revised the text to clearly address that this study performed in older adults and rationales was also about older adults. I also added the limitation of generalizability of this study

before

after

Depressive symptoms experienced during the COVID-19 pandemic have been reported to be caused by social isolation.

Depressive symptoms experienced during the COVID-19 pandemic have been reported to be caused by social isolation in older adults.

Depressive symptoms can be indirectly affected by social isolation, as well as directly through the mediating effect of loneliness.

Depressive symptoms can be indirectly affected by social isolation, as well as directly through the mediating effect of loneliness in older adults [4].

This study examined the effects of social isolation due to social distancing on depressive symptoms and the mediating effect of loneliness. Additionally, the moderating effects of moderate and mild physical activity on depressive symptoms were influenced by social isolation and loneliness.

This study examined the effects of social isolation due to social distancing on depressive symptoms and the mediating effect of loneliness in older adults. Additionally, the moderating effects of moderate and mild physical activity on depressive symptoms were influenced by social isolation and loneliness in older adults.

The results revealed that the impact of social isolation on depressive symptoms is directly and indirectly mediated by loneliness.

The results revealed that the impact of social isolation on depressive symptoms is directly and indirectly mediated by loneliness in older adults.

This study also identified the effects of physical activity on the mediation model.

This study also identified the effects of physical activity on the mediation model in older adults.

Finally, this study was conducted using a general sample from the elderly population. Separate studies of patients with depression and healthy populations should be conducted to establish a deeper understanding of depressive symptoms in older adults.

Finally, this study was conducted using a general sample from the elderly population in England. Additional studies with other population from various cultures would be necessary to generalize our findings. In addition, separate studies of patients with depression and healthy populations should be conducted to establish a deeper understanding of depressive symptoms in older adults.

      Question 6: The paper indicates that depressive symptoms are a significant issue for older adults during the COVID-19 pandemic due to increased social isolation. A potential issue might be whether the paper sufficiently establishes the novelty of its contribution to the existing literature.

   Answer 6: To our knowledge, this is the first paper examining social isolation due to the social distancing during the pandemic, loneliness, depressive symptoms, and physical activities in older adults experiencing COVID-19 altogether as one comprehensive moderated mediation model. I agree that this study is not enough to find out all potential variables and issues regarding depressive symptoms of older adults during the pandemic, this paper could draw attention and act as a stepping stone for further investigation.

      Question 6: The paper outlines its main objectives, but it might be critiqued for not stating specific hypotheses or research questions, which could provide clarity and focus for the reader.

   Answer 6: Thank you for your advice. I added specific hypotheses in the end of the introduction section as you have recommended.

before

after

-

Hypotheses of this study were: (1) the higher social isolation score would be associated with the high level of depressive symptoms (direct path), (2) the social isolation score would be associated with the higher loneliness score, and the higher loneliness score would be associated with the higher the high level of depressive symptoms (indirect path of mediation), (3) physical activity would moderate the effect of social isolation score and loneliness on depressive symptoms, which could mean that individuals with the higher level of physical activity would experience lower level of depressive symptoms (moderated mediation), and (4) moderating effect of physical activity would be different among the severity of activity.

Materials and methods

     Question 7: Please provide an example of item for the scales adopted in the study.

   Answer 7: Thank you for your suggestion. I included examples of items used in this study.

before

after

Depression (CES-D-8) scale [21]. The CES-D-8 consists of 8 items with a dichotomous response format.

Depression (CES-D-8) scale [23]. The CES-D-8 consists of 8 items with a dichotomous response format, including questions such as “Much of the time during the past week: You felt depressed” and “Much of the time during the past week: felt sad” for example.

Social isolation during the COVID-19 pandemic was ascertained using two items: self-isolation in the previous week and trying to stay at home during the last week.

Social isolation during the COVID-19 pandemic was ascertained using two items: self-isolation in the previous week and trying to stay at home during the last week.

Loneliness was assessed using the University of California, Los Angeles Loneliness Scale-Short Form (ULS-4) [22]. The ULS-4 consists of 4 items and is assessed on a 3-point Likert scale ranging from 1 (“hardly ever or never”) to 3 (“often”).

Loneliness was assessed using the University of California, Los Angeles Loneliness Scale-Short Form (ULS-4) [24]. The ULS-4 consists of 4 items including questions such as “How often do you feel you lack companionship?” and “How often do you feel lonely?” for example. It is assessed on a 3-point Likert scale ranging from 1 (“hardly ever or never”) to 3 (“often”).

     Question 8:  Please specify direction of the correlation between gender and depression.

   Answer 8: Thank you for the detailed feedback. I added the direction in the text.

before

after

gender (r = .158, p < .001),

gender (r = .158, p < .001; significant correlation between female and more depressive symptom),

      Question 9: Please describe the nature of the relationship between age and depression.

   Answer 9: Thank you for the detailed feedback. I added the direction in the text.

before

after

age (r = -.047, p < .001),

age (r = -.047, p < .001; significant correlation between lesser age and more depressive symptom),

     Question 10:  Could you elucidate on the coding schema employed for the variable of ethnicity? Is it not classified as a categorical variable?

   Answer 10: I deeply appreciate your keen advice. I added all the coding information of covariates of this study from the ELSA COVID-19 dataset.

before

after

-

Demographic variables and general health variables from the existing datasets were set as covariates including gender (1 = male; 2 = female), age (years), ethnicity (1 = Any White; 2 = Black, Asian and minority ethnic), relationship status (1 = Married; 2 = In a registered civil partnership; 3 = Living with a partner; 4 = With a partner you do not live with; 5 = Separated (after being married or in a civil partnership); 6 = Divorced/dissolved civil partnership; 7 = Widowed/surviving partner from a civil partnership; 8 = Single), tenure (1 = Own it outright; 2 = Buying it with the help of a mortgage or loan; 3 = Pay part rent and part mortgage (shared ownership); 4 = Rent it; 5 = Live here rent free (including rent free in relative's / friend's property; excluding squatting); 6 = Squatting), smoking (1 = Yes; 2 = No), weight (in kilograms), height (in centimeters), perceived health status of the last month (1 = Excellent; 2 = Very good; 3 = Good; 4 = Fair; 5 = Poor), perceived sleep status of the last month (1 = Excellent; 2 = Very good; 3 = Good; 4 = Fair; 5 = Poor), and having long-standing illness, disability, or infirmity (1 = Yes; 2 = No).

      Question 11: Many of the correlations denoted as significant may be exaggerated due to the sample size. The magnitude of the effect size for the associations between depressive symptoms and factors such as social isolation, loneliness, moderately energetic physical activity, and mildly energetic physical activity appears to be minimal.

   Answer 11: I understand that correlation seems to be weak in some cases. I aware that these weak but statistically significant associations could be induced due to the large sample size. However, in regression analyses, paths of those small correlation coefficients also showed significant regression weight. Moreover, previous studies also reported significant relationships between social isolation and depressive symptoms (Elmer & Stadtfeld, 2020; Ge et al., 2017; Robb et al., 2020; Santini et al., 2020), between social isolation and loneliness (Banerjee, 2020; Ge et al., 2017; Hsiao et al., 2023; Robb et al., 2020), and between physical activity and depressive symptoms (Bursnall, 2014; Ku et al., 2018; Overdorf et al., 2016; Pickett et al., 2012). Therefore, results of this study including weak but statistically significant associations are in line with many other previous studies. However, I agree with your opinion and added this information as limitation of this study.

Banerjee, D. (2020). The impact of Covid‐19 pandemic on elderly mental health. International journal of geriatric psychiatry, 35(12), 1466.

Bursnall, P. (2014). The relationship between physical activity and depressive symptoms in adolescents: a systematic review. Worldviews on Evidence‐Based Nursing, 11(6), 376-382.

Elmer, T., & Stadtfeld, C. (2020). Depressive symptoms are associated with social isolation in face-to-face interaction networks. Scientific reports, 10(1), 1444.

Ge, L., Yap, C. W., Ong, R., & Heng, B. H. (2017). Social isolation, loneliness and their relationships with depressive symptoms: A population-based study. PloS one, 12(8), e0182145.

Hsiao, F. Y., Peng, L. N., Lee, W. J., & Chen, L. K. (2023). Sex-specific impacts of social isolation on loneliness, depressive symptoms, cognitive impairment, and biomarkers: Results from the social environment and biomarker of aging study. Archives of Gerontology and Geriatrics, 106, 104872.

Ku, P. W., Steptoe, A., Liao, Y., Sun, W. J., & Chen, L. J. (2018). Prospective relationship between objectively measured light physical activity and depressive symptoms in later life. International journal of geriatric psychiatry, 33(1), 58-65.

Overdorf, V., Kollia, B., Makarec, K., & Alleva Szeles, C. (2016). The relationship between physical activity and depressive symptoms in healthy older women. Gerontology and geriatric medicine, 2, 2333721415626859.

Pickett, K., Yardley, L., & Kendrick, T. (2012). Physical activity and depression: A multiple mediation analysis. Mental Health and Physical Activity, 5(2), 125-134.

Robb, C. E., De Jager, C. A., Ahmadi-Abhari, S., Giannakopoulou, P., Udeh-Momoh, C., McKeand, J., ... & Middleton, L. (2020). Associations of social isolation with anxiety and depression during the early COVID-19 pandemic: a survey of older adults in London, UK. Frontiers in psychiatry, 11, 591120.

Santini, Z. I., Jose, P. E., Cornwell, E. Y., Koyanagi, A., Nielsen, L., Hinrichsen, C., ... & Koushede, V. (2020). Social disconnectedness, perceived isolation, and symptoms of depression and anxiety among older Americans (NSHAP): a longitudinal mediation analysis. The Lancet Public Health, 5(1), e62-e70.

before

after

-

In addition, some correlation coefficients and mediated moderation indices were statistically significant but weak due to the large sample size. Although correlations and moderating effects were not only significant but also in line with previous studies, further studies with smaller samples could reduce the possible error regarding the sample size.

      Question 12: Social isolation was ascertained using two binary response items. This operational definition may not capture the full spectrum of social isolation experiences

Answer 12: I understand your advice. This study operationally defines social isolation as an objective state of social distancing, such as self-isolation and trying to stay at home during the COVID-19 pandemic. This definition might not cover all the meaning and spectrum of social isolation and I stated this point as the limitation of this study already. However, there is no study focused on social isolation due to the social distancing and lockdown of the pandemic yet. I strongly think that this information would be significant if a similar pandemic occurs in the future.

before

after

First, it operationally defines social isolation as an objective state of social distancing, such as self-isolation and trying to stay at home during the COVID-19 pandemic. However, social isolation could include other perspectives, such as the absence of contact with other people or even a subjective feeling of lack of belongingness [47, 28]. Although this study focused on social isolation caused by social distancing due to the pandemic, further studies including various components of social isolation should be conducted to improve our understanding.

First, it operationally defines social isolation as an objective state of social distancing, such as self-isolation and trying to stay at home during the COVID-19 pandemic. However, social isolation could include other perspectives, such as the absence of contact with other people or even a subjective feeling of lack of belongingness [51, 52]. Although this study focused on social isolation caused by social distancing due to the pandemic, further studies including various components of social isolation and compare them before and after COVID-19 pandemic should be conducted to improve our understanding.

      Question 13: The sentence implies that the analyses are driven by data rather than by theory, as would be appropriate. “Therefore, moderate and mild physical activities could be set as possible moderating variables because they showed a significant correlation with the dependent variable, whereas they did not show a significant correlation with the independent or mediating variables. However, vigorous physical activity was not set as a possible moderating variable because it did not show a significant correlation with the dependent variable.”

   Answer 13: The sentence you pointed out meant that hypothesized regression model could be analyzed due to the significant correlation results in my data prior to the regression analyses. To my knowledge, this sort of sentence is commonly used to describe the general process of regression analyses. 

Results

      Question 14: Upon examination of the correlation matrix, the sole discernible significant effect appears to be the positive association between loneliness and depression, alongside those among the different modalities of physical activity.

   Answer 14: I understand that correlation seems to be weak in some cases. I aware that these weak but statistically significant associations could be induced due to the large sample size. However, in regression analyses, paths of those small correlation coefficients also showed significant regression weight. Moreover, previous studies also reported significant relationships between social isolation and depressive symptoms (Elmer & Stadtfeld, 2020; Ge et al., 2017; Robb et al., 2020; Santini et al., 2020), between social isolation and loneliness (Banerjee, 2020; Ge et al., 2017; Hsiao et al., 2023; Robb et al., 2020), and between physical activity and depressive symptoms (Bursnall, 2014; Ku et al., 2018; Overdorf et al., 2016; Pickett et al., 2012). Therefore, results of this study including weak but statistically significant associations are in line with many other previous studies. However, I agree with your opinion and added this information as limitation of this study.

Banerjee, D. (2020). The impact of Covid‐19 pandemic on elderly mental health. International journal of geriatric psychiatry, 35(12), 1466.

Bursnall, P. (2014). The relationship between physical activity and depressive symptoms in adolescents: a systematic review. Worldviews on Evidence‐Based Nursing, 11(6), 376-382.

Elmer, T., & Stadtfeld, C. (2020). Depressive symptoms are associated with social isolation in face-to-face interaction networks. Scientific reports, 10(1), 1444.

Ge, L., Yap, C. W., Ong, R., & Heng, B. H. (2017). Social isolation, loneliness and their relationships with depressive symptoms: A population-based study. PloS one, 12(8), e0182145.

Hsiao, F. Y., Peng, L. N., Lee, W. J., & Chen, L. K. (2023). Sex-specific impacts of social isolation on loneliness, depressive symptoms, cognitive impairment, and biomarkers: Results from the social environment and biomarker of aging study. Archives of Gerontology and Geriatrics, 106, 104872.

Ku, P. W., Steptoe, A., Liao, Y., Sun, W. J., & Chen, L. J. (2018). Prospective relationship between objectively measured light physical activity and depressive symptoms in later life. International journal of geriatric psychiatry, 33(1), 58-65.

Overdorf, V., Kollia, B., Makarec, K., & Alleva Szeles, C. (2016). The relationship between physical activity and depressive symptoms in healthy older women. Gerontology and geriatric medicine, 2, 2333721415626859.

Pickett, K., Yardley, L., & Kendrick, T. (2012). Physical activity and depression: A multiple mediation analysis. Mental Health and Physical Activity, 5(2), 125-134.

Robb, C. E., De Jager, C. A., Ahmadi-Abhari, S., Giannakopoulou, P., Udeh-Momoh, C., McKeand, J., ... & Middleton, L. (2020). Associations of social isolation with anxiety and depression during the early COVID-19 pandemic: a survey of older adults in London, UK. Frontiers in psychiatry, 11, 591120.

Santini, Z. I., Jose, P. E., Cornwell, E. Y., Koyanagi, A., Nielsen, L., Hinrichsen, C., ... & Koushede, V. (2020). Social disconnectedness, perceived isolation, and symptoms of depression and anxiety among older Americans (NSHAP): a longitudinal mediation analysis. The Lancet Public Health, 5(1), e62-e70.

before

after

-

In addition, some correlation coefficients and mediated moderation indices were statistically significant but weak due to the large sample size. Although correlations and moderating effects were not only significant but also in line with previous studies, further studies with smaller samples could reduce the possible error regarding the sample size.

      Question 15: Please provide the beta coefficients for the regression models to ascertain the magnitude of the effect.

   Answer 15: The PROCESS macro by Dr. Hayse does not provide standardized regression coefficients other than mediation-only model. Standardized regression coefficients can be generated manually by standardizing all variables prior to the use of the PROCESS. However, reporting a standardized coefficient was not recommended since the bootstrap confidence intervals from PROCESS should not be interpreted as confidence intervals for the standardized effects, for that is not what they are (from FAQ page of PROCESS by Dr. Hayse; https://processmacro.org/faq.html). Therefore, I would like to report unstandardized regression coefficients only, as the PROCESS macro reports out.

      Question 16: Why was Model 15 employed instead of other models, such as Model 7 or 8? Please explicate the theoretical rationale underpinning this decision.

   Answer 16: Thank you for your advice. I added the rationale of this decision in the text as you have recommended.

before

after

Daily moderate and moderate-to-vigorous physical activity were negatively associated with depression among older adults during COVID-19 [19].

Daily moderate and moderate-to-vigorous physical activity were negatively associated with depression among older adults during COVID-19 [19]. In addition, physical activity was a significantly relevant variable regarding relationship between social activity and depression [17, 20] and between loneliness and depression [21].

A moderated mediation model from PROCESS macro number 15 was used to discover the mediating effect of loneliness on the relationship between social isolation and depressive symptoms, and the moderating effect of moderate and mild physical activities on depressive symptoms

A moderated mediation model from PROCESS macro number 15 was used to discover the mediating effect of loneliness on the relationship between social isolation and depressive symptoms, and the moderating effect of moderate and mild physical activities of effects of both social isolation and loneliness on depressive symptoms

      Question 17: The mediated moderation index (β = -.004, CI [-.008, -.001]) and moderation effects were reported as significant. However, these effects appear to be extremely small.

   Answer 17: I understand that mediated moderation effect seems to be weak. However, as you have mentioned, both mediated moderation index and regression weights of the moderation are statistically significant. Therefore, I believe that these are still meaningful results. However, I agree with your opinion and added this information as limitation of this study.

before

after

-

In addition, some correlation coefficients and mediated moderation indices were statistically significant but weak due to the large sample size. Although correlations and moderating effects were not only significant but also in line with previous studies, further studies with smaller samples could reduce the possible error regarding the sample size.

     Question 18:  Please, verify the accuracy of the reported R².

   Answer 18: The PROCESS macro only generates R, R², MSE, F, df1, df2, and p score for model summary. I have reported R², F, and p score in tables previously, and I added R and MSE in the revised version. In addition, I have already written the changes in R² and F scores due to moderation effects in the text

      Question 19: The study controls for several covariates, but it is not clear how the selection of these covariates was determined, or whether additional variables could have influenced the results.

   Answer 19: Thank you for your feedback. The socio-demographic variables including gender, age, ethnicity, tenure, and socioeconomic status as money insufficiency, lifestyle variables including current smoking status, and health variables including self-reported chronic diseases and BMI were commonly used as covariates in previous studies regarding social isolation, loneliness, physical activity, and depressive symptoms in older adults (Ge et al., 2017; Lee et al., 2014). I added the rationale in the text as you recommended.

before

after

Demographic variables, including gender, age, ethnicity, relationship status, tenure, and socioeconomic status, and general health variables, including smoking, weight, height, body mass index (BMI), perceived health status of the last month, perceived sleep status of the last month, and having long-standing illness, disability, or infirmity, were used from the datasets to control for possible contaminating effects.

Demographic variables, including gender, age, ethnicity, relationship status, and tenure as an index of socioeconomic status, and general health variables, including smoking, weight, height, body mass index (BMI), perceived health status of the last month, perceived sleep status of the last month, and having long-standing illness, disability, or infirmity, were used from the datasets to control for possible contaminating effects according to previous studies regarding social isolation, loneliness, physical activity, and depressive symptoms in older adults [21, 22].

      Question 20: The paper excluded individuals with missing values, which could introduce bias if the missing data are not random.

Answer 20: I understand your point. However, out of a total of 6794 people, only 296 were excluded due to missing values, which is only about 4%. I believe that it would not be problematic to run the analysis with a total of 6499 people.

Discussion

Question 21: The study defines social isolation in terms of social distancing behaviors like self-isolation and staying at home. This definition may not capture the full extent or impact of social isolation on mental health, which can include aspects such as the absence of meaningful social relationships or a subjective feeling of loneliness.

Answer 21: Thank you for your keen advice. As I stated above, this definition might not cover all the meaning and spectrum of social isolation and I stated this point as the limitation of this study already. However, there is no study focused on social isolation due to the social distancing and lockdown of the pandemic yet. I strongly think that this information would be significant if a similar pandemic occurs in the future.

before

after

First, it operationally defines social isolation as an objective state of social distancing, such as self-isolation and trying to stay at home during the COVID-19 pandemic. However, social isolation could include other perspectives, such as the absence of contact with other people or even a subjective feeling of lack of belongingness [47, 28]. Although this study focused on social isolation caused by social distancing due to the pandemic, further studies including various components of social isolation should be conducted to improve our understanding.

First, it operationally defines social isolation as an objective state of social distancing, such as self-isolation and trying to stay at home during the COVID-19 pandemic. However, social isolation could include other perspectives, such as the absence of contact with other people or even a subjective feeling of lack of belongingness [51, 52]. Although this study focused on social isolation caused by social distancing due to the pandemic, further studies including various components of social isolation and compare them before and after COVID-19 pandemic should be conducted to improve our understanding.

Question 22: The study’s cross-sectional design limits its ability to establish causality between the variables of interest. Longitudinal studies would be more informative in terms of how changes over time in social isolation, loneliness, and physical activity relate to changes in depressive symptoms.

Answer 22: I deeply agree with your comment. I have already stated my concern about this cross-sectional design as the limitation of my study. My initial aim of this study was to examine the longitudinal effect of social isolation at Wave 1 on depressive symptoms at Wave 2. However, Wave 1 data were collected in June and July 2020, the earlier stage of the pandemic, and might not reflect the problems caused by the COVID-19 lockdown. I will examine the longitudinal effect after releasing new data in future. In addition, I will look for variables related to social isolation other than social distancing due to the COVID-19 lockdown used in both pre-pandemic and post-pandemic dataset and compare them.

before

after

Although this study focused on social isolation caused by social distancing due to the pandemic, further studies including various components of social isolation should be conducted to improve our understanding. Moreover, this study was performed with a cross-sectional design using wave 2 data of ELSA COVID-19 alone. This design was internally made since Wave 2 data were collected in November and December 2020, while Wave 1 data were collected in June and July 2020, the earlier stage of the pandemic, and might not reflect the problems caused by the COVID-19 lockdown. Further longitudinal studies should be conducted after releasing new data.

Although this study focused on social isolation caused by social distancing due to the pandemic, further studies including various components of social isolation and compare them before and after COVID-19 pandemic should be conducted to improve our understanding. Moreover, this study was performed with a cross-sectional design using wave 2 data of ELSA COVID-19 alone. This design was internally made since Wave 2 data were collected in November and December 2020, while Wave 1 data were collected in June and July 2020, the earlier stage of the pandemic, and might not reflect the problems caused by the COVID-19 lockdown. Further longitudinal studies should be conducted after releasing new data.

Question 23: The study's findings are based on data from older adults in England during a specific period of the COVID-19 pandemic. These findings may not be generalizable to other populations, times, or contexts.

Answer 23: I agree with your comments. I added this information as the limitation of this study.

before

after

Finally, this study was conducted using a general sample from the elderly population. Separate studies of patients with depression and healthy populations should be conducted to establish a deeper understanding of depressive symptoms in older adults.

Finally, this study was conducted using a general sample from the elderly population in England. Additional studies with other population from various cultures would be necessary to generalize our findings. In addition, separate studies of patients with depression and healthy populations should be conducted to establish a deeper understanding of depressive symptoms in older adults.

      Question 24: The study concludes that moderate to mild physical activity can reduce depressive symptoms influenced by social isolation and loneliness. However, the discussion acknowledges that physical activity does not have a significant relationship with social isolation or loneliness. This might suggest the need for exploring other factors that could mediate or moderate the relationship between these variables.

   Answer 24: Thank you for your keen advice. Physical activity did not show significant relations with social isolation or loneliness in this study, which made it possible to be set as moderator since moderating variables should not correlated with independent variables. However, I agree that future studies on exploring the exact relationship among physical activity, social isolation, and loneliness is necessary and should be performed with careful consideration of possible mediators or moderators. I added this information in Discussion section.

before

after

On the other hand, results showed insignificant relationship of physical activity with social isolation and loneliness, which support previous studies that physical activity did not show significant relationship with social isolation or loneliness [42, 43, 44]. Although some studies have reported contradictory outcomes suggesting a significant relationship between physical activity, social isolation, and loneliness [17, 45, 46], not all studies have found a direct negative relationship between physical activity and social isolation or loneliness, which raises the possibility of other moderators and mediators within this relationship [46]. Thus, future studies should include other underlying variables to identify the relationship between physical activity and social isolation or loneliness.

On the other hand, results showed insignificant relationship of physical activity with social isolation and loneliness, which support previous studies that physical activity did not show significant relationship with social isolation or loneliness [48, 49, 50]. It was possible to set physical activity as the moderator in this study since moderating variables should not correlated with independent variables [52]. Although some studies have reported contradictory outcomes suggesting a significant relationship between physical activity, social isolation, and loneliness [17, 52, 53], not all studies have found a direct negative relationship between physical activity and social isolation or loneliness, which raises the possibility of other moderators and mediators within this relationship [52]. Thus, future studies should include other underlying variables to identify the relationship between physical activity and social isolation or loneliness with careful consideration of possible mediators or moderators.

      Question 25: The study discusses the potential applicability of its findings to future pandemics. While this is an important consideration, the unique characteristics of each pandemic situation need to be taken into account when applying these findings.

   Answer 25: I understand about your concern. Although each pandemic situation has its unique characteristics, mental health problems with social isolation issue could occur. In fact, MERS, SARS and COVID-19 share similar features including highly contagious zoonotic viruses and are all related to mental health problems including depression. Therefore, I strongly think that results of this study would be significant if a similar pandemic occurs in the future.

before

after

While COVID-19 has become endemic, other infectious diseases may emerge and cause significant adverse effects on mental health, such as Middle East respiratory syndrome (MERS) and severe acute respiratory syndrome (SARS) once were before COVID-19. The results of this study could serve as a stepping stone to understand the activating and buffering effects of depression in the elderly during the next possible pandemic.

While COVID-19 has become endemic, other infectious diseases may emerge and cause significant adverse effects on mental health, such as Middle East respiratory syndrome (MERS) and severe acute respiratory syndrome (SARS) once were before COVID-19. MERS, SARS and COVID-19 share similar features including highly contagious zoonotic viruses [51] and are all related to mental health problems including depression [52, 53]. The results of this study could serve as a stepping stone to understand the activating and buffering effects of depression in the elderly during the next possible pandemic.

Question 26: Please, may the authors elucidate how these findings correspond with those derived from the subsequent paper, and identify the elements that theoretically and empirically distinguish the two studies. https://www.mdpi.com/1660-4601/20/19/6900

Answer 26: The author of the paper you have mentioned is the corresponding author of this paper. In our previous study, we found that both social isolation and loneliness were associated with depressive symptoms, but that loneliness was more strongly associated with depressive symptoms than social isolation. Furthermore, these findings were examined in young adults. In order to further this knowledge and explore its applicability to other populations, we found previous studies reporting the mediating effect of the loneliness between social isolation and depressive symptoms in older adults. Therefore, we elucidated this mediation model in this 

Reviewer 2 Report

Comments and Suggestions for Authors

This study examined the effect of social isolation on depression as mediated by loneliness, and moderated by physical activity, during the covid-19 pandemic. Social isolation is a typical phenomenon during the pandemic, and it is significant to examine how it could affect depression. I think the research question is important and the research was rigorous. I recommend the paper for publication. 

Only one suggestion: The physical activity act as a buffer to the deleterious effect of both social isolation and loneliness in the authors’ model, but the literature review merely focuses on how physical activities are associated with depression. It could add to the depth of the review by adding some discussion on the potential buffering effect of physical activity. 

Author Response

Reviewer #2

This study examined the effect of social isolation on depression as mediated by loneliness, and moderated by physical activity, during the covid-19 pandemic. Social isolation is a typical phenomenon during the pandemic, and it is significant to examine how it could affect depression. I think the research question is important and the research was rigorous. I recommend the paper for publication. 

Only one suggestion: The physical activity act as a buffer to the deleterious effect of both social isolation and loneliness in the authors’ model, but the literature review merely focuses on how physical activities are associated with depression. It could add to the depth of the review by adding some discussion on the potential buffering effect of physical activity. 

Answer: Thank you for understanding and empathizing with the implications of my research. I also appreciate your keen advice. I added the interpretation of the moderating effect of physical activity on depression in Discussion. 

before

after

Behavioral activation is targeted at reducing behavioral inertia and avoidance, which often accompanies depression, by increasing daily activities [40]. A recent study also suggested that physical activity is an important component and logical basis of behavioral activation interventions for depression [41]. Therefore, improving physical activity could act as a potential therapeutic component of behavioral activation for interventions in depressive symptoms.

Behavioral activation is targeted at reducing behavioral inertia and avoidance, which often accompanies depression, by increasing daily activities [42]. A recent study also suggested that physical activity is an important component and logical basis of behavioral activation interventions for depression [43]. Opportunities to engage in physical activity during the COVID-19 pandemic was impaired due to the lockdown and financial insecurities [18]. That is, engaging in physical activity regularly during the pandemic might establish individuals' health competence and utilize physical activity as a coping strategy [44, 45]. Therefore, improving physical activity could act as a potential therapeutic component of behavioral activation for interventions in depressive symptoms.

Reviewer 3 Report

Comments and Suggestions for Authors

dear

This is a very well written article and the analysis is good. However, your study is simply analysis. Request a more advanced analysis.

I would like to make two major changes.

1. Are there any differences in the figures for social isolation, loneliness, physical activity, and depressive symptoms in 2020 compared to the time when COVID-19 did not exist?

2. If you add the odds ratio through logistic regression analysis, you will be able to see the differences in social isolation, loneliness, and depressive symptoms according to physical activity. Or, conversely, analysis may be possible.

There is a reason why research related to COVID-19 is important to us. For this reason, if a similar pandemic occurs in the future, it is intended to reduce negative factors such as social isolation, loneliness, and depressive symptoms for the elderly and to further improve health and wellness.

Comments on the Quality of English Language

I have no special opinion.

Author Response

Reviewer #3

This is a very well written article and the analysis is good. However, your study is simply analysis. Request a more advanced analysis.

 I would like to make two major changes.

Question 1: Are there any differences in the figures for social isolation, loneliness, physical activity, and depressive symptoms in 2020 compared to the time when COVID-19 did not exist?

Answer 1: Thank you for your suggestion. This study operationally defines social isolation as an objective state of social distancing, such as self-isolation and trying to stay at home during the COVID-19 pandemic. Therefore, there is no variable that equally measured social isolation in pre-pandemic data. However, I strongly think that your suggestion is important, and your suggestion has inspired me for further research. In a future study, I will look for variables related to social isolation other than social distancing due to the COVID-19 lockdown used in both pre-pandemic and post-pandemic and compare them.

before

after

Although this study focused on social isolation caused by social distancing due to the pandemic, further studies including various components of social isolation should be conducted to improve our understanding.

Although this study focused on social isolation caused by social distancing due to the pandemic, further studies including various components of social isolation and compare them before and after COVID-19 pandemic should be conducted to improve our understanding.

Question 2: If you add the odds ratio through logistic regression analysis, you will be able to see the differences in social isolation, loneliness, and depressive symptoms according to physical activity. Or, conversely, analysis may be possible.

Answer 2: The logistic regression is known to be performed in situations where a linear relationship cannot be assumed, i.e., when the dependent variable is categorical. In this paper, all variables, including physical activity, are continuous, so a linear relationship can be assumed. It would be more appropriate to proceed with correlation and regression analysis to determine the linear relationship between variables. In addition, in this paper, physical activity was not significantly related to social isolation or loneliness, but was only significantly related to depressive symptom, which is described in the results section. I believe that the correlation and regression analysis are sufficient to fulfill the purpose of this paper, but please let me know if additional analyses related to physical activity seems to be necessary.

There is a reason why research related to COVID-19 is important to us. For this reason, if a similar pandemic occurs in the future, it is intended to reduce negative factors such as social isolation, loneliness, and depressive symptoms for the elderly and to further improve health and wellness.

Round 2

Reviewer 1 Report

Comments and Suggestions for Authors

Dear authors, 

I commend your efforts in revising the manuscript in response to the feedback provided during the initial review. The enhancements made have undoubtedly strengthened the overall quality of your work, underscoring your commitment to addressing feedback and elevating the scholarly contribution of your research.  

However, upon re-evaluating the revised manuscript, I identified two remaining concerns that I believe merit further attention and consideration.

Similarity with Previous Work:

It appears that the issue of similarity with a prior paper by the same author persists. While I acknowledge your efforts to address this concern, I suggest additional revisions to more distinctly differentiate the current work from your previous publication. Providing a clearer delineation of the unique contributions and expanding the discussion on the novel aspects introduced in this manuscript would be beneficial.

Relative Weakness of Effect Sizes:

The second concern pertains to the relatively weak effect sizes observed in the results. Although you have made progress in addressing this issue, I recommend providing a more in-depth discussion on the implications of these effect sizes. Consider discussing potential limitations and offering insights into the practical significance of your findings. Additionally, suggesting future research directions to strengthen the overall impact of the study would be valuable.

 Comments on the Quality of English Language

The overall quality of the English is sufficient, with only a few minor issues that need checking.

Author Response

Response to Reviewers’ Comments

Reviewer #1

Similarity with Previous Work:

Question 1: It appears that the issue of similarity with a prior paper by the same author persists. While I acknowledge your efforts to address this concern, I suggest additional revisions to more distinctly differentiate the current work from your previous publication. Providing a clearer delineation of the unique contributions and expanding the discussion on the novel aspects introduced in this manuscript would be beneficial.

Answer 1: Thank you for your advice. I added the clearer statement of the unique contribution of this study in discussion section as you recommended.

before

after

This study is significant because it explored depressive symptoms using a moderated mediation model of social isolation, loneliness, and physical activity in older adults during the COVID-19 period. The results of this study provide evidence for the knowledge of depressive symptoms, with the activating effect of social isolation directly and indirectly mediated by loneliness, and with the buffering effect of moderate and mild physical activities of older adults during a pandemic.

This study is significant because it explored depressive symptoms using a moderated mediation model of social isolation, loneliness, and physical activity in older adults during the COVID-19 period. Previous studies focused on discovering relationships among social isolation, loneliness and depressive symptoms [23, 54, 55], particularly the mediating effect of loneliness between social isolation and depressive symptoms [4, 14, 15]. However, there is no study yet to discover the possible moderation effect of physical activity with the mediating effect. To our knowledge, this study is the first study to examine the comprehensive moderated mediation model of social isolation, loneliness, depressive symptoms, and physical activity altogether in older adults. The results of this study provide evidence for the knowledge of depressive symptoms, with the activating effect of social isolation directly and indirectly mediated by loneliness, and with the buffering effect of moderate and mild physical activities of older adults during a pandemic.

Relative Weakness of Effect Sizes:

Question 2: The second concern pertains to the relatively weak effect sizes observed in the results. Although you have made progress in addressing this issue, I recommend providing a more in-depth discussion on the implications of these effect sizes. Consider discussing potential limitations and offering insights into the practical significance of your findings. Additionally, suggesting future research directions to strengthen the overall impact of the study would be valuable.

Answer 2: Thank you for your advice. I agree that careful discussion is needed to enlarge the implication of this study. I added the more description about weak but significant results in discussion section as you recommended.

before

after

In addition, some correlation coefficients and mediated moderation indices were statistically significant but weak due to the large sample size. Although correlations and moderating effects were not only significant but also in line with previous studies, further studies with smaller samples could reduce the possible error regarding the sample size.

In addition, some correlation coefficients and mediated moderation indices were statistically significant but weak due to the large sample size. However, the large sample size resembles the true population [61], so that coefficients fluctuate less which makes results meaningful enough. In fact, weak but significant results of this study were in line with previous studies. To discover the exact and reliable results, further studies with smaller samples could reduce the possible error regarding the sample size.

Reviewer 3 Report

Comments and Suggestions for Authors

Thanks for the revision. I hope for good results.

Comments on the Quality of English Language

I have no comment.

Author Response

Thank you for your review.